# Impact of the COVID-19 pandemic and a supertyphoon: A quantitative study in Cebu, Philippines

**Michelle Ylade[1], Bipin Adhikari[2,3]\*, Maria Vinna Crisostomo[1], Jedas Veronical Daag[1], Anna Maureen Cuachin[1], March Helena Lopez[1], Angela Macasero[1], Kristal An Agrupis[1], Jacqueline Deen[1]**

**1** Institute of Child Health and Human Development National Institutes of Health University of the Philippines Manila, Manila, Philippines, **2** Mahidol Oxford Tropical Medicine Research Unit, Faculty of Tropical Medicine, Mahidol University, Bangkok, Thailand, **3** Centre for Tropical Medicine and Global Health, Nuffield Department of Medicine, University of Oxford, Oxford, United Kingdom

\* Bipin@tropmedres.ac

**Data Availability Statement:** The dataset that informed the results in this manuscript are available at an online data repository, Harvard

## Abstract

Pandemics and natural disasters are recognized to cause major disruptions. The main objective of this study was to explore the impacts of COVID-19 and supertyphoon Odette in Cebu, Philippines. A total of 2630 participants were interviewed exploring the impacts of COVID-19 and supertyphoon Odette. The majority of the respondents (2486/2630; 94.5%) had financial problems due to COVID-19. Almost three out of four respondents (1962/2630; 74.6%) experienced moderate to severe impact on their mental health. Almost a third of the respondents (874/2630; 33.2%) reported moderate to severe impact on their physical well-being, mostly related to weight-related disorders. Almost half of the respondents (1248/2630; 47.5%) experienced moderate to severe impacts on their relationships with family members, relatives, friends and neighbors. More than two-thirds of the respondents (1673/2360; 63.6%) reported moderate to severe financial problems due to supertyphoon Odette. Households who were financially impacted by Supertyphoon Odette were more likely not have recently migrated to their current residence (p<0.001), and to have lower monthly expenditure (p = 0.020). The specific financial problems reported by the majority (1671/2360; 64.5%) were increased expenses, followed by inability to work (623/2360; 23.7%). Almost two-thirds of the respondents (1680/2360; 63.9%) reported having mental health problems. The majority of respondents (1853/2360; 70.5%) had moderate to severe impacts on their living conditions, disrupted by interruption in electricity, water supplies, and house damage. The COVID-19 pandemic and supertyphoon Odette had multi-faceted effects with immediate and long-term implications and greater impacts among poorer households. Public health measures to counteract the consequences of both of these incidents require a multi-pronged and targeted approach.

Dataverse at following URL: https://doi.org/10.7910/DVN/UZAK3D.

**Funding:** This study received financial support from the Philippines Department of Health. The funders had no role in study design, data collection and analysis, decision to publish, or preparation of the manuscript.

**Competing interests:** The authors declare that they have no competing interests. BA serves as an academic editor of PLOS Global Public Health

**Abbreviations:** COVID-19, Corona virus Disease 2019; USA, United States of America; USD, : United States Dollar.

## Introduction

COVID-19 was a major pandemic and has already caused more than 700 million morbidity and 7 million deaths to date, and significant proportion of population continues to suffer from disabilities [1]. While the severity of infection caused significant scores of morbidities and mortalities during the peaks of the pandemic and drained substantial resources, the myriad consequences after the pandemic are rarely accounted [2]. COVID-19 also had major impacts on the global economy but the impacts among low- and middle-income countries (LMICs) were higher because of the pre-existing fragile health system, and vulnerability of the population [3–5].

A state of public health emergency due to COVID-19 was declared in the Philippines in March 2020 and lifted in July 2023. During this period, COVID-19 is estimated to have caused over 4 million cases and 66,864 deaths in the country [6] and was associated with a Gross Domestic Product (GDP) contraction in 2020 of -9.5% [7]. The pandemic also exposed the fragile health system of the Philippines, reducing access to health services and treatment of routine health conditions, which led to an increase in non-communicable diseases and mental health conditions [8].

Natural disasters are insurmountable challenges to human beings and continue to claim a huge toll of mortality and morbidity throughout the history of human civilization. Globally, between 1994 and 2013, there were 6,873 natural disasters globally which claimed 1.35 million lives that translated to a loss of 68,000 lives on average each year affecting 218 million people during this 20 year period [9]. Natural disasters also affect LMICs disproportionately, for instance, higher-income countries experienced 56% of disasters but lost 32% of lives, while lower-income countries experienced 44% of disasters and suffered 68% of deaths [9]. The direct economic losses caused by storm disasters in 2021 were the largest, reaching 137.7 billion USD [10]. The Asia pacific region is considered susceptible to increasing number of disasters in recent years. For instance, between 2014 and 2017, the Asia pacific regions was affected by 55 earthquakes, 217 storms and cyclones, and 236 cases of severe flooding, impacting 650 million people and causing 33,000 deaths [11].

The Philippines, comprising three major island groups (Luzon, Visayas and Mindanao), is considered to be highly prone to natural hazards. About 60% of its land area and 74% of its population is at risk of floods, cyclones, droughts, earthquakes, tsunamis and landslides [12]. Since 1990, there have been an estimated 565 large natural disasters in the country that killed about 70,000 people and produced an economic loss of 23 billion USD [12]. During the COVID-19 pandemic, on 16th December 2021, superyphoon Odette made landfall several times on various islands in Visayas and Mindanao producing torrential rains, violent winds, landslides, and storm surges which affected over 7.8 million people and damaged one out of every three hospitals (28%) [13].

The impacts of COVID-19 and Supertyphoon Odette are multifaceted, affecting populations in both the short and long term [14, 15]. These effects range from individual physical and mental well-being to broader social disruptions, such as employment, social cohesion, school attendance, power outages, and housing damage [16]. Exploring these impacts is critical to informing future preparedness against pandemic and disasters relevant to specific social, cultural and political context [5, 17]. Previous studies have explored the effects of COVID-19 and supertyphoon Odette on Filipinos, but they were limited by their narrow focus and small sample sizes [18–23]. The main objective of this study was to explore the multifaceted impact of COVID-19 and supertyphoon Odette on Filipino adolescents and their corresponding household head, parent, or guardian.

## Materials and methods

### Settings

This study was conducted in Bogo city and Balamban municipality, both located in Cebu province, one of the 82 provinces in the Philippines (**Fig 1**). Cebu is situated in the central Visayas region and has a population of 3,325,385, based on the census of 2020 [24]. Bogo city is located in the northeastern coast of Cebu and has a population of 88,867 people [25]. On 8th November 2013, 95% of Bogo city was destroyed by typhoon Haiyan and is considered to be the most powerful typhoon ever to make landfall in recorded history [26, 27]. The storm surges were responsible for 6300 deaths, 1061 missing and injuring 28689 people in the country [26]. Balamban is a municipality located on the western coast of Cebu and has a population of 95,136 people [28]. The municipality is an industrial economic hub, and notably hosts multinational shipbuilding corporations and is recognized as a 'shipbuilding capital of the Philippines' [28].

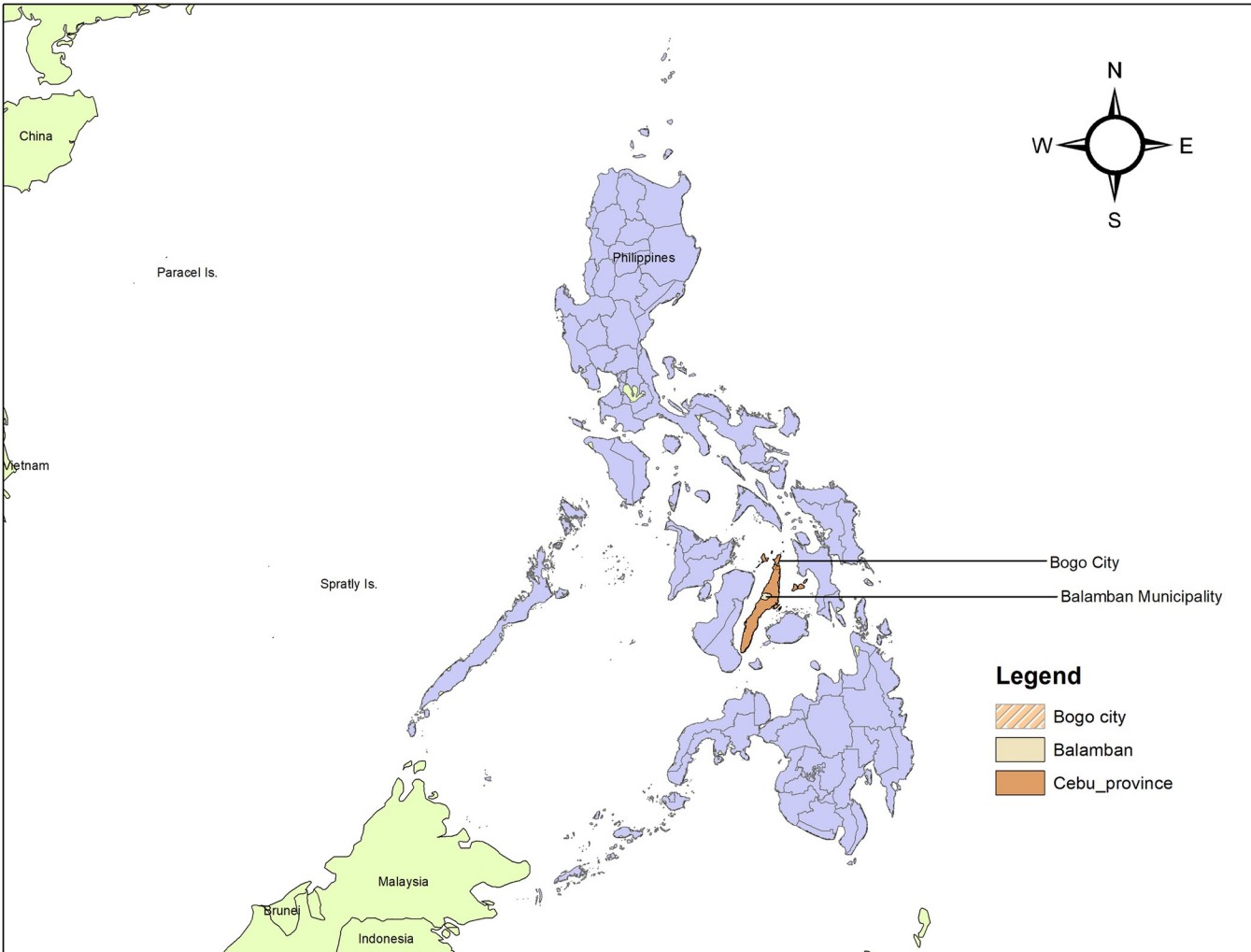

**Fig 1. Map of the Philippines showing the island of Cebu and the study sites in Balamban municipality and Bogo city.** The map was created by the authors in ESRI's ArcGIS 10.3.1. Shapefiles for provincial and district boundaries were obtained from the Humanitarian Data Exchange https://data.humdata.org/dataset/cod-ab-phl and are licensed under a Creative Commons Attribution 4.0 (CC-BY 4.0) International license.

## Study context

This study was embedded in a longitudinal, prospective, population-based cohort study that evaluated the relative risk of developing virologically confirmed dengue among those who did or did not receive a single dose of CYD-TDV by previous dengue virus infections at baseline (Clinicaltrials.gov, NCT03465254) [29]. This cross-sectional survey was built on the sample from the cohort study, where a total of 2996 healthy children (aged 9–14 years) residing in Bogo or Balamban. These children were enrolled between May and June 2017, and were followed up for acute febrile illness until October 2022.

## Study design

This study was a quantitative questionnaire-based survey amongst the guardians of children who participated in the dengue cohort study. The study followed a STROBE checklist for a cross-sectional study (S1 Text).

## Participants

Rural Health Units prepared a master list with the names, dates of birth, sexes and addresses of healthy children aged 9 to 14 years residing in Cebu, who were invited to participate in the vaccination campaign. The master list included 285,242 children, of whom 149,023 (52.2%) received a dose of CYD-TDV from 15 June to 16 August 2017.

Between July 2022 and June 2023, we interviewed the children's corresponding household heads or parent/guardians using a standard questionnaire (S2 Text) to evaluate the impact of the COVID-19 pandemic and other disasters on the participants' social, economic, physical, and mental well-being.

## Data collection

The questionnaire was designed to explore the impact of COVID-19 and supertyphoon Odette among the cohort of our participants [30]. The questionnaire was developed based on the questions informed by the previously used instrument including literature that explored the impact of COVID-19 [31–36] and disaster on human health [37–41]. The final set of questions in the questionnaire was tested among the authors first, followed by seeking feedback from the social scientist (BA) and were refined for the pre-test. The questionnaire was further refined based on the feedback from the initial sub-set of participants and was launched finally among the participants.

The first section of the questionnaire collected brief identifiers of the respondent including the date and time of interview. The subsequent section collected data on socio-economic and environmental profile of the household. For instance, that included the building material of the house, number of household members, and their duration of residence including brief demographic, socio-behavioural and economic variables.

The final two sections each explored the impact of COVID-19 and typhoon Odette on households with multiple options, with the severity of impacts rated on a Likert scale from the mildest (score = 0) and to the strongest (score = 10). The impacts due to COVID-19 and typhoon Odette are outcome variables and are linked to our research question: 'What are the impacts of COVID-19 and typhoon Odette on Filipino adolescents and their corresponding household head or parent/guardian?'

Impacts for instance were assessed in terms of disruption in workability, employment, salary, expenses including option for free text. Mental health impacts were assessed by variables such as anxiety, depression, agitation, and frustration. Impact on physical well-being was assessed in terms of weight gain/loss, COVID-19 infection itself, and co-infections/co-morbid

conditions. Similarly impact on their social conditions were assessed by exploring their social contact, fights/disagreements, perceived inferiority/lack of achievement and any other specific disruptions. Number of household members who caught COVID-19 and any mortalities from the infection were also assessed. Similar to the impact of COVID-19, for supertyphoon Odette, three major domains were explored that included disruption in financial problems, mental health, and living conditions.

The responses were recorded on paper during face-to-face interviews and entered into an electronic data capture system that was developed for the study.

### Data analysis

The primary methods of data analysis were descriptive statistics, informed by the study objectives, to explore the impacts of COVID-19 and the supertyphoon. The analysis included both descriptive and inferential statistics. The descriptive analysis involved calculating percentages, with proportions categorized into each domain. The responses were categorized on an ascending-order of Likert scale, where 0 indicated the least impact and 10 indicated the highest impact. The inferential analysis included exploring tests for associations between categorical variables using Fisher's exact and Chi-squared tests, and the Mann–Whitney U test for comparing discrete variables between classifications of categorical variables. Characteristics of those who were impacted and those who were not impacted by COVID-19 and Supertyphoon Odette were compared by category. Respondents who scored zero (0) in the different categories were classified as "not impacted", and those who scored at least 1 were classified as "impacted". A p-value of ≤0.05 was considered statistically significant. Data were analyzed using STATA (version 18), Stata statistical software, college station, Texas: Stata Corp LLC.

### Ethical approval

The study protocol was approved by the University of the Philippines–Manila Research Ethics Board. A parent or legal guardian of the participants provided written informed consent. Verbal assent was obtained from the participants and documented (**S3 Text**).

## Results

### Overview

From July 2022 to June 2023, we interviewed 2630 (78.8%) of the 2996 participants originally enrolled into the study in 2017 and their corresponding household head or parents/guardian. We assessed the demographic, socio-behavioral, and economic characteristics of the participants (**S1 Table**). The mean age of the adolescent participants at the start of the qualitative assessment was 16.4 years and ranged from 14.2 to 20.9 years. About half of the participants lived in houses composed of cement (1334/2630; 50.7%) and the rest lived in wooden houses. The majority of the adolescent participants (2,591 98.5%) lived in households with access to electricity. The median number of household members was six. The majority of the household heads (95.3%; 2506/2630) had more than 6 years of school education. Nearly all households (98.4%; 2587/2630) had mobile phones; and more than half (57.1%; 1502/2630) had motorbikes. More than two-thirds of the households had monthly expenditures between 5000 to 10000 Php.

### Impact of COVID-19 on the participants' household heads or parents/guardians

Among the several domains of assessment that were examined through the survey, financial problems had the greatest impact. The majority of the respondents (2486/2630; 94.5%) shared

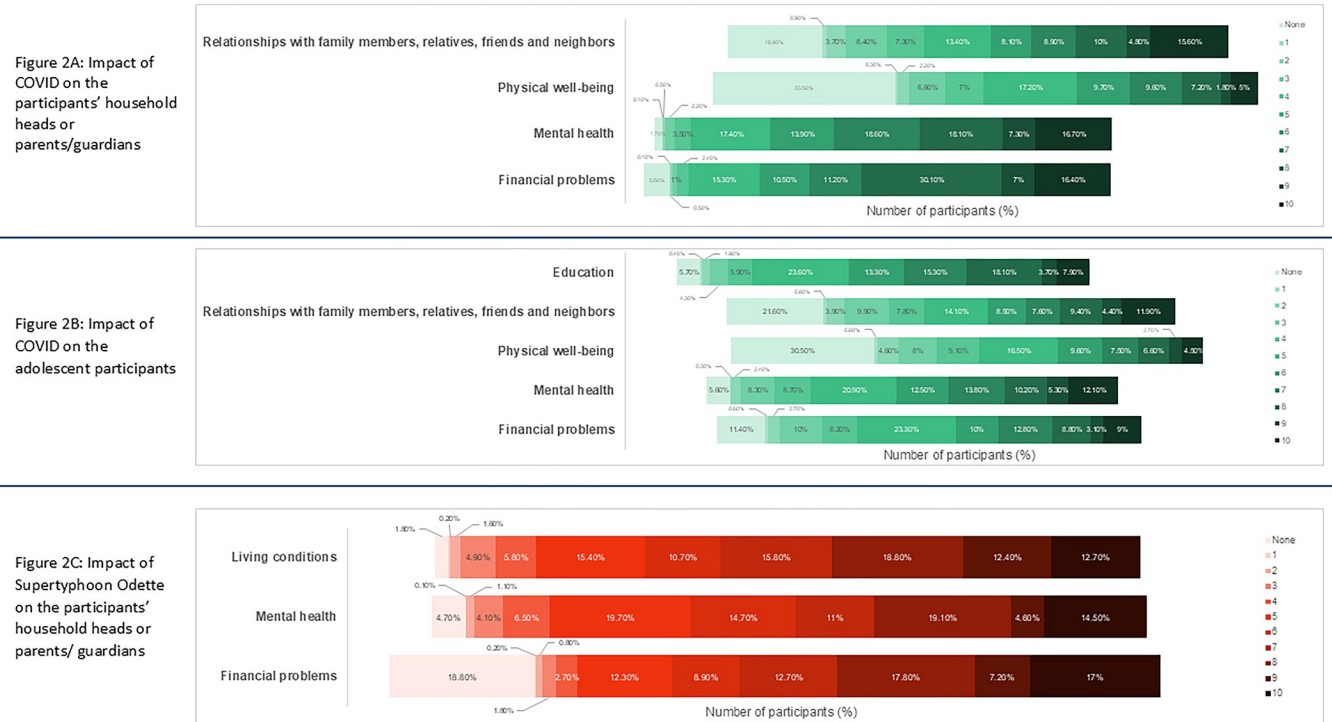

**Fig 2. Impact of COVID-19 and supertyphoon on the participants' household heads or parents/guardians.**

that they had financial problems due to COVID-19 (**S2 Table**). The majority of respondents (1978/2630; 75.2%) had moderate to severe financial impacts based on the self-reported scores that ranged between 6 to 10 (**Fig 2**). More than two-thirds (1825/2630; 69.4%) incurred increased expenses due to COVID-19, and two-fifths of the respondents (1100/2630; 41.8%) were unable to work, almost half (487/1100; 44.2%) reported having to suspend their work for one month.

We compared the characteristics of households financially impacted by COVID-19 with those that were not (**Table 1**). Households that were financially impacted were more likely to be from Bogo (p<0.001), to have recently migrated to their current residence (p = 0.048), and to have lower estimated monthly expenditures (p<0.001). COVID-19 had a significant impact on mental health. Nearly three out of every four respondents (1962/2630; 74.6%) experienced a moderate to severe impact on their mental health. Among these respondents, a significant majority (2223/2630; 84.5%) had symptoms related to anxiety and almost half (1259/2630; 47.9%) had depression. About a third of the respondents (874/2630; 33.2%) reported moderate to severe impact on their physical well-being. More than two thirds developed weight related disorders; among these one third of the respondents gained weight (875/2630; 33.3%) and another one third lost weight (847/2630; 32.2%). About 40% (908/2630) of the respondents had multiple co-morbidities such as hypertension, stroke, and arthritis.

There were significantly more households from Balamban whose physical well-being was impacted compared to those from Bogo (p<0.001). Additionally, households whose physical well-being were impacted were more likely to have household head with >6 years of schooling (p<0.001) and to not have recently migrated to their current residence (p = 0.026).

Almost half of the respondents (1248/2630; 47.5%) had moderate to severe impact on relationship with family members, relatives, friends and neighbors. Majority of the respondents

**Table 1. Impacts of COVID-19 to the household.**

| Domain-I: Financial impacts | | | |
|---|---|---|---|
| **Characteristics** | **Financially impacted (n, %)** | **Not financially impacted (n, %)** | **p-value** |
| | **2,486 (94.5)** | **144 (5.5)** | |
| **Place of residence** | | | |
| Bogo | 1,298 (52.2) | 45 (31.2) | <0.001* |
| Balamban | 1,188 (47.8) | 99 (68.8) | |
| **Numbers of individual in the household** | | | |
| Mean (SD) | 6 (2) | 6 (2) | 0.579¶ |
| Median (IQR) | 5 (4–7) | 5 (4–7) | |
| **Number of children within household** | | | |
| Mean (SD) | 3 (1) | 3 (1) | 0.647¶ |
| Median (IQR) | 3 (2–3) | 3 (2–3) | |
| **Household head with >6 years schooling** | 2,371 (95.3) | 135 (93.8) | 0.414* |
| **Migrated to current residence in the past 2 years** | 64 (2.6) | 0 (0.0) | 0.048** |
| **Has access to electricity** | 2,447 (98.4) | 144 (100.0) | 0.271* |
| **Estimated monthly household expenditure** | | | |
| Mean | 11,437.1 (6,202.6) | 14,284.7 (6,660.0) | <0.001¶ |
| Median (IQR) | 10,000 (8,000–15,000) | 12,000 (10,000–20,000) | |
| Domain-II: Physical wellbeing impacts | | | |
| **Characteristics** | **Physical well-being impacted (n, %)** | **Physical well-being not impacted (n, %)** | **p-value** |
| | **1,750 (66.5)** | **498** | |
| **Place of residence** | | | |
| Bogo | 776 (44.3) | 567 (64.4) | <0.001* |
| Balamban | 974 (55.7) | 313 (35.6) | |
| **Numbers of individual in the household** | | | |
| Mean (SD) | 6 (2) | 6 (2) | 0.566¶ |
| Median (IQR) | 5 (2–14) | 5 (2–14) | |
| **Number of children within household** | | | |
| Mean (SD) | 3 (1.4) | 3 (1.5) | 0.054¶ |
| Median (IQR) | 2 (2–3) | 2 (2–3) | |
| **Household head with >6 years schooling** | 1,695 (96.9) | 811 (92.2) | <0.001* |
| **Migrated to current residence in the past 2 years** | 35 (2.0) | 29 (3.3) | 0.042** |
| **Has access to electricity** | 1,726 (98.6) | 865 (98.3) | 0.498* |
| **Estimated monthly household expenditure** | | | |
| Mean | 12,039.5 (6,494.8) | 10,705.2 (5,667.1) | <0.001¶ |
| Median (IQR) | 10,000 (2,000–60,000) | 10,000 (3,000–40,000) | |
| Domain-III: Relational impacts | | | |
| **Characteristics** | **Relational well-being impacted (n, %)** | **Relational well-being not impacted (n, %)** | **p-value** |
| | **2,132 (81.1)** | **498 (18.9)** | |
| **Place of residence** | | | |
| Bogo | 1,111 (52.1) | 232 (46.6) | 0.026* |
| Balamban | 1,021 (47.9) | 266 (53.4) | |
| **Numbers of individual in the household** | | | |
| Mean (SD) | 6 (2) | 6 (2) | 0.523¶ |
| Median (IQR) | 5 (4–7) | 5 (4–7) | |
| **Number of children within household** | | | |
| Mean (SD) | 3 (1) | 3 (2) | 0.631¶ |

*(Continued)*

**Table 1.** (Continued)

| Domain-I: Financial impacts | | | |
|---|---|---|---|
| Median (IQR) | 2 (2–3) | 3 (2–3) | |
| **Household head with >6 years schooling** | 2,056 (96.4) | 450 (90.4) | <0.001* |
| **Migrated to current residence in the past 2 years** | 45 (2.1) | 19 (3.8) | 0.035** |
| **Has access to electricity** | 2,101 (98.6) | 490 (98.4) | 0.836* |
| **Estimated monthly household expenditure** | | | |
| Mean | 11,740.3 (6396.8) | 10,962.4 (5603.4) | 0.002¶ |
| Median (IQR) | 10,000 (8,000–15,000) | 10,000 (7,500–13,000) | |

*Chi-squared test for categorical variables with cell sizes more than 50

**Fisher Exact test for categorical variables with cell sizes less than 50

¶Mann Whitney U test for median values between two categories.

(2024/2630; 77%) reported a decrease in social contact. Households whose relational well-being was impacted were more likely to be from Bogo (p = 0.026), have a household head with more than 6 years of schooling (p<0.001), not have recently migrated to their current residence (p = 0.035), and have higher monthly expenditures (p = 0.002).

## Impact of COVID-19 to the adolescent participants

Almost two-thirds of respondents had moderate to severe detrimental effects on their education because of COVID-19 pandemic. Majority of the respondents had difficulty with online work (2069/2630; 78.7%), and almost two-fifth (1133/2630; 43.1%) respondents missed the company of their classmates (S3 Table). There were no significant differences between participants whose schooling was impacted by COVID-19 and those who were not, except that those who were not impacted were significantly more likely to have recently migrated to their current residence (p = 0.026) (Table 2).

More than two-fifths of the respondents reported severe financial problems (1148/2360; 43.7%) and majority of the respondents (1983/2360; 75.4%) reported the reasons for such financial problems to be lack of allowance or inability to work (Fig 2). Participants who were financially impacted by COVID-19 were more likely to be from Balamban (p<0.001), belong to a household with a head who had more than 6 years of schooling (p = 0.002), and have access to electricity (p = 0.037).

More than half of the respondents (1416/2360; 53.8%) reported having moderate to severe forms of mental health problems. The respondents reported being bored (1819/2360; 69.2%), anxious (1275/2360; 48.5%) and depressed (635/2360; 24.1%). Participants whose mental health was impacted by COVID-19 were significantly more likely to have a household head with more than 6 years of schooling (p<0.001). Conversely, residents of Bogo were more likely to report that their mental health was not impacted (p<0.001).

Almost one third of the respondents (811/2360; 30.8%) had moderate to severe impacts on their physical well-being. Most of these respondents' physical well-being were related to weight disorders; the majority (1120/2360; 42.6%) reported gaining weight in comparison to losing weight (704/2360; 26.8%). Participants whose physical well-being was impacted by COVID-19 were significantly more likely to have a household head with more than 6 years of schooling (p = 0.001) and to report higher estimated monthly household expenditure (p<0.001). In contrast, residents of Bogo were more likely to report that their physical well-being was not impacted (p<0.001).

**Table 2. Impacts of COVID-19 to the cohort participant.**

| Domain-I: Schooling impacts | | | |
|---|---|---|---|
| Characteristics | Schooling impacted (n, %) | Schooling not impacted (n, %) | p-value |
| | 2,481 (94.3) | 149 (5.7) | |
| Place of residence | | | |
| Bogo | 1,266 (51.0) | 77 (48.3) | 0.877* |
| Balamban | 1,215 (49.0) | 72 (51.7) | |
| Numbers of individual in the household | | | |
| Mean (SD) | 6 (2) | 6 (2) | 0.550¶ |
| Median (IQR) | 5 (4–7) | 5 (4–7) | |
| Number of children within household | | | |
| Mean (SD) | 3 (1) | 3 (2) | 0.454¶ |
| Median (IQR) | 2 (2–3) | 2 (1–3) | |
| Household head with >6 years schooling | 2,365 (95.3) | 141 (94.6) | 0.689* |
| Migrated to current residence in the past 2 years | 56 (2.3) | 8 (5.4) | 0.026** |
| Has access to electricity | 2,443 (98.5) | 148 (99.3) | 0.723* |
| Estimated monthly household expenditure | | | |
| Mean | 11,619.6 (6,273.3) | 11,151.9 (6,050.9) | 0.128¶ |
| Median (IQR) | 10,000 (8,000–15,000) | 10,000 (8,000–12,000) | |
| Domain-II: Financial impacts | | | |
| Characteristics | Financially impacted (n, %) | Financially not impacted (n, %) | p-value |
| | 2,329 (88.6) | 301 (11.4) | |
| Place of residence | | | |
| Bogo | 1,099 (47.2) | 244 (81.1) | <0.001* |
| Balamban | 1,230 (52.8) | 57 (18.9) | |
| Numbers of individual in the household | | | |
| Mean (SD) | 6 (2) | 6 (2.5) | 0.039¶ |
| Median (IQR) | 5 (4–7) | 6 (4–7) | |
| Number of children within household | | | |
| Mean (SD) | 3 (1) | 3 (2) | 0.162¶ |
| Median (IQR) | 2 (2–3) | 2 (1–3) | |
| Household head with >6 years schooling | 2,230 (95.6) | 276 (91.7) | 0.002* |
| Migrated to current residence in the past 2 years | 53 (2.3) | 11 (3.7) | 0.161** |
| Has access to electricity | 2,299 (98.7) | 292 (97.0) | 0.037* |
| Estimated monthly household expenditure | | | |
| Mean | 11,599.4 (6,195.5) | 11,543.6 (6,755.1) | 0.109¶ |
| Median (IQR) | 10,000 (8,000–15,000) | 10,000 (7,500–15,000) | |
| Domain-III: Mental health impacts | | | |
| Characteristics | Mental health impacted (n, %) | Mental health not impacted (n, %) | p-value |
| | 2,484 (94.4) | 146 (5.6) | |
| Place of residence | | | |
| Bogo | 1,240 (49.9) | 103 (70.6) | <0.001* |
| Balamban | 1,244 (50.1) | 43 (29.4) | |
| Numbers of individual in the household | | | |
| Mean (SD) | 6 (2) | 6 (2) | 0.991¶ |
| Median (IQR) | 5 (4.5–7) | 5 (4–7) | |
| Number of children within household | | | |
| Mean (SD) | 3 (1.5) | 3 (2) | 0.428¶ |
| Median (IQR) | 2 (2–3) | 2 (2–3) | |

*(Continued)*

**Table 2.** (Continued)

| | Household head with >6 years schooling col1 | col2 | p-value |
|---|---|---|---|
| Household head with >6 years schooling | 2,377 (95.7) | 129 (88.4) | <0.001* |
| Migrated to current residence in the past 2 years | 60 (2.4) | 4 (2.7) | 0.779** |
| Has access to electricity | 2,449 (98.6) | 142 (97.3) | 0.168* |
| Estimated monthly household expenditure | | | |
| Mean | 11,589.1 (6,113.6) | 11,659.7 (8,404.9) | 0.139¶ |
| Median (IQR) | 10,000 (8,000–15,000) | 10,000 (7,000–15,000) | |
| **Domain-IV: Physical well-being impacts** | | | |
| **Characteristics** | **Physical well-being impacted (n, %)** | **Physical well-being not impacted (n, %)** | **p-value** |
| | **1,829 (69.5)** | **801 (30.5)** | |
| **Place of residence** | | | |
| Bogo | 837 (45.8) | 506 (63.2) | <0.001* |
| Balamban | 992 (54.2) | 295 (36.8) | |
| **Numbers of individual in the household** | | | |
| Mean (SD) | 6 (2) | 6 (2) | 0.390¶ |
| Median (IQR) | 5 (4–7) | 6 (5–7) | |
| **Number of children within household** | | | 0.090¶ |
| Mean (SD) | 3 (1.5) | 3 (1.5) | |
| Median (IQR) | 2 (2–3) | 2 (2–3) | |
| **Household head with >6 years schooling** | 1,760 (96.2) | 746 (93.1) | 0.001* |
| **Migrated to current residence in the past 2 years** | 41 (2.2) | 23 (2.9) | 0.335** |
| **Has access to electricity** | 1,802 (98.5) | 789 (98.5) | 1.000* |
| **Estimated monthly household expenditure** | | | |
| Mean | 11,850.6 (6,387.4) | 11,004.9 (5,923.4) | <0.001¶ |
| Median (IQR) | 10,000 (8,000–15,000) | 10,000 (7,000–15,000) | |
| **Domain-V: Relational impacts** | | | |
| **Characteristics** | **Relational well-being impacted (n, %)** | **Relational well-being not impacted (n, %)** | **p-value** |
| | **2,061 (78.4)** | **569 (21.6)** | |
| **Place of residence** | | | |
| Bogo | 1,053 (51.1) | 290 (51.0) | 0.958* |
| Balamban | 1,008 (48.9) | 279 (49.0) | |
| **Numbers of individual in the household** | | | |
| Mean (SD) | 6 (2) | 6 (2) | 0.180¶ |
| Median (IQR) | 5 (4–7) | 6 (4–7) | |
| **Number of children within household** | | | |
| Mean (SD) | 3 (1) | 3 (1.5) | 0.584¶ |
| Median (IQR) | 2 (2–3) | 2 (2–4) | |
| **Household head with >6 years schooling** | 1,988 (96.5) | 518 (91.0) | <0.001* |
| **Migrated to current residence in the past 2 years** | 48 (2.3) | 16 (2.8) | 0.538** |
| **Has access to electricity** | 2,029 (98.4) | 562 (98.8) | 0.697* |
| **Estimated monthly household expenditure** | | | |
| Mean | 11,652.2 (6210.6) | 11,379.0 (6,440.3) | 0.084¶ |
| Median (IQR) | 10,000 (8,000–15,000) | 10,000 (8,000–15,000) | |

*Chi-squared test for categorical variables with cell sizes more than 50

**Fisher Exact test for categorical variables with cell sizes less than 50

¶Mann Whitney U test for median values between two categories.

A considerable proportion of the respondents (1101/2630; 41.9%) had moderate to severe impact on relationship with family members, relatives, friends and neighbors. Majority of the respondents (1985/2630; 75.5%) reported a decrease in social contact. There were no significant differences between participants whose relational well-being was impacted by COVID-19 and those who were not, except that participants whose well-being was impacted were more likely to belong to a household where the head had more than 6 years of schooling (p<0.001).

## Impact of supertyphoon Odette on the participants' household heads or parents/guardians

Almost two-third of the respondents (1673/2360; 63.6%) reported moderate to severe financial problems (**Fig 2**). The specific financial problems reported by the majority (1671/2360; 64.5%) were an increase in expenses, followed by inability to work (623/2360; 23.7%) (**S4 Table**). Households who were financially impacted by Supertyphoon Odette were more likely from Balamban (p<0.001), not have recently migrated to their current residence (p<0.001), and had lower monthly expenditures (p = 0.020) (**Table 3**). Around two-thirds of the respondents (1680/2360; 63.9%) reported having mental health problems. The majority reported experiencing anxiety (2099/2360; 79.8%), boredom (890/2360; 33.8%) and depression (888/2360; 33.8%) as major mental health issues.

The majority of respondents (1853/2360; 70.5%) had moderate to severe impacts on their living conditions. These conditions were disrupted by interruption in electricity, water supplies, house damage. The vast majority of respondents (2566/2360; 97.6%) reported loss of electricity, with an average duration of 20 days [range 0–120 days]. A significant proportion also experienced disrupted water supply; the average duration of water supply disruption was 13 days [range 0–90 days]. The majority (1034/2360; 96.9%) of respondents experienced partial damage to their dwellings, with significant damage reported among 40% of the respondents (1066/2360). Residents of Bogo were more likely to report that their mental health were not impacted by Supertyphoon Odette (p<0.001).

## Discussion

COVID-19 and supertyphoon Odette affected the quality of life of the adolescent participants and their corresponding household heads or parents/guardians through various ways. The majority of respondents explained that both of these disasters increased financial problems, worsened mental health, deteriorated physical well-being, and strained relationships with family members and friends. The inferential analysis showed greater vulnerability among poorer households with more severe impacts across all domains (financial, physical, mental). Supertyphoon Odette also caused significant house damage and prolonged disruptions in electricity, internet and water supply. The co-occurrence of COVID-19 and the supertyphoon exacerbated the impact and added an extra layer of burden to population and the health system [42–44]. COVID-19 was characterized by community lockdowns and stay-at-home orders, which were difficult to comply with under normal circumstances but became nearly impossible when there was physical damage to houses and extended interruptions in electricity, internet and water supply [45]. Both the supertyphoon and the pandemic can have long-term impacts on individuals, with financial and mental health consequences leading to a poor quality of life [46].

### Impact of COVID-19

There was a similar impact of COVID-19 on the adolescent participants as on their household heads or parents/guardians, and the major effect was on their financial situation. Poorer

**Table 3. Impacts of supertyphoon Odette to the household.**

| Domain-I: Financial impacts | | | |
|---|---|---|---|
| **Characteristics** | **Financially impacted (n, %)** | **Financially not impacted (n, %)** | **p-value** |
| | **2,137 (81.3)** | **493 (18.7)** | |
| **Place of residence** | | | |
| Bogo | 936 (43.8) | 407 (82.6) | <0.001* |
| Balamban | 1,201 (56.2) | 86 (17.4) | |
| **Numbers of individual in the household** | | | |
| Mean (SD) | 6 (2) | 6 (2) | 0.326¶ |
| Median (IQR) | 5 (5–7) | 5 (4–7) | |
| **Number of children within household** | | | |
| Mean (SD) | 3 (1) | 3 (2) | 0.006¶ |
| Median (IQR) | 2 (2–3) | 2 (1–3) | |
| **Household head with >6 years schooling** | 2,039 (95.4) | 467 (94.7) | 0.516* |
| **Migrated to current residence in the past 2 years** | 32 (1.5) | 32 (6.5) | <0.001** |
| **Has access to electricity** | 2,106 (98.6) | 485 (98.4) | 0.836* |
| **Estimated monthly household expenditure** | | | |
| Mean | 11,389.0 (5,934.3) | 12,477.7 (7,456.2) | 0.020¶ |
| Median (IQR) | 10,000 (8,000–15,000) | 10,000 (8,000–15,000) | |
| Domain-II: Mental health impacts | | | |
| **Characteristics** | **Mental health impacted (n, %)** | **Mental health not impacted (n, %)** | **p-value** |
| | **2,506 (95.3)** | **124 (4.7)** | |
| **Place of residence** | | | |
| Bogo | 1,228 (49.0) | 115 (92.7) | <0.001* |
| Balamban | 1,278 (51.0) | 9 (7.3) | |
| **Numbers of individual in the household** | | | |
| Mean (SD) | 6 (2) | 6 (2.5) | 0.491¶ |
| Median (IQR) | 5 (4–7) | 5 (4–6) | |
| **Number of children within household** | | | |
| Mean (SD) | 3 (1.5) | 3 (2) | 0.463¶ |
| Median (IQR) | 2 (2–3) | 2 (2–3) | |
| **Household head with >6 years schooling** | 2,388 (95.3) | 118 (95.2) | 0.830* |
| **Migrated to current residence in the past 2 years** | 60 (2.4) | 4 (3.2) | 0.542** |
| **Has access to electricity** | 2,474 (98.7) | 117 (94.3) | 0.002* |
| **Estimated monthly household expenditure** | | | |
| Mean | 11,608.2 (6,250.4) | 11,286.3 (6,485.8) | 0.154¶ |
| Median (IQR) | 10,000 (8,000–15,000) | 10,000 (7,000–15,000) | |

*Chi-squared test for categorical variables with cell sizes more than 50

**Fisher Exact test for categorical variables with cell sizes less than 50

¶Mann Whitney U test for median values between two categories.

households had higher detrimental impacts by the pandemic and echoes with a study from Thailand which showed that more than 70% of the households experienced a decline in income, and around 60% of the low-income households experienced deprivation of food during the COVID-19 pandemic [47]. The global economy was estimated to lose nearly of 8.5 trillion USD between 2020 to 2024, with 34.3 million people being pushed below the poverty line in 2020 [48].

Loss of jobs during the COVID-19 pandemic was a prominent finding of our study and highlights how the pandemic affected the employment landscape. According to data from the

US Bureau of Labor Statistics, employment in the USA declined by 9.4 million USD in 2020, the largest calendar-year decline in history [49]. In our study almost half of the participants had to suspend their work for a month. Our findings support those from a World Bank report of severe job losses among Filipino construction workers (56%), public transportation workers (52%), agricultural workers (70%) and other small-scale workers, with the brunt of employment loss continuing to affect the population despite gradual recovery [50]. Losing jobs and employment opportunities can have multi-dimension impacts and are recognized causes of anxiety and depression [51].

Even when there are no direct impacts of COVID-19 due to illness or death, the pandemic itself can have subtle and indirect impacts affecting mental health [52]. In this study, three out of four participants experienced significant impact on mental health, with anxiety being the most common complaint. The fear of the pandemic and its potential to cause illness and death may be considered as more detrimental than COVID-19 itself [53, 54].

COVID-19 also impacts on the physical well-being and may manifest through somatic symptoms such as pain, weight related disorders and increased vulnerability to other illnesses [55, 56]. Almost half of our respondents were found to gain weight and one-fifth were found to lose weight during the pandemic [57]. Lockdowns and travel restrictions have been found to have major impacts on health and well-being of the population, sometimes leading to extreme events such as suicide and violence [58–60].

COVID-19 was also found to disrupt the social sphere, interpersonal bonding and social capital, primarily because of its detrimental effects on social distancing, communication gaps, and other barriers [61]. Contiguity, physical presence, and social relationship are deemed one of the critical behavioral elements of human civilization in fostering attachment and good mental health among the individuals [62]. The majority of the respondents in this study experienced diminished social contacts.

Households in Bogo experienced greater financial impacts from the COVID-19 pandemic compared to those in Balamban. This may be attributed to the direct and immediate impact of COVID-19 on the financial reliance of urban Bogo on trade and employment, in contrast to the more rural and still agriculture-based Balamban municipality.

## Impact of supertyphoon Odette

In this study, more than two-thirds of the respondents reported having mental health problems, and most reported anxiety, and depression. Disasters are recognized to have aggravated the mental health problems across the globe [63]. For instance, 12 years after the Hurricane Katrina, post-traumatic stress disorder continues to affect one in six mothers from low-income household [64]. This study demonstrated greater impacts among poorer households across all domains, highlighting how disasters can increase vulnerabilities. Disasters can push the populations into a vicious cycle of poverty leading to mental health problems, and mental health problems subsequently intensifying poverty, deepening the inequities in health care access, education, employment and living [65].

Importantly, the infrastructural damages due to supertyphoons can have catastrophic consequences such as immediate loss of living space, disruption of electricity, and water supplies and subsequent damage to living and livelihood of residents [66, 67]. Physical damage due to disasters can result into a multi-faceted disruptions apart from significant loss of economy, by escalating further vulnerability among population who could be deprived from access to health care, income generating activities, and other unforeseen risks [68, 69]. The households in Balamban were heavily impacted by supertyphoon Odette because of the typhoon's path, with the strongest winds sweeping through the central part of Cebu province, including Balamban.

### Strengths and limitations

This study was embedded in a cohort study and thus leveraged the large sample size available for the survey. Long-term and ongoing contact between study staff and the respondents fostered trust and encouraged truthful responses to the questions. The studied variables offered a great detail on which domains the pandemic and the natural disaster affected the respondents, including the magnitude of impact, although the reasons and mechanisms on how those domains were affected were beyond the scope of this study. In the future, qualitative studies including focus group discussions and in-depth interviews could potentially delve deeper into how these domains were affected including the narratives around the themes and their development.

### Conclusions

Disasters such as the COVID-19 pandemic and supertyphoons can have severe consequences and have multi-faceted impacts with both immediate and long-term implications. Both of these incidents affected the financial situation, mental health, physical well-being and social conditions of the respondents. Unlike disease epidemics, supertyphoons are typically abrupt and disrupt physical infrastructure, which can lead to longer-term effects on mental health, and social and familial well-being. Both supertyphoons and the COVID-19 pandemic significantly impacted most domains of human health, with greater effects observed among poorer households. This highlights the need for targeted strategies to support vulnerable households.

### Supporting information

**S1 Text. STROBE checklist for cross-sectional study.**
(PDF)

**S2 Text. Questionnaire used to interview respondents in this study.**
(PDF)

**S3 Text. Inclusivity in global health.**
(PDF)

**S1 Table. Socio-demographics of the participants.**
(DOCX)

**S2 Table. Impacts of COVID-19 on households.**
(DOCX)

**S3 Table. Impacts of COVID-19 on participants.**
(DOCX)

**S4 Table. Impacts of supertyphoon.**
(DOCX)

### Author Contributions

**Conceptualization:** Michelle Ylade, Maria Vinna Crisostomo, Jedas Veronical Daag, Anna Maureen Cuachin, March Helena Lopez, Angela Macasero, Kristal An Agrupis, Jacqueline Deen.

**Data curation:** Michelle Ylade, Maria Vinna Crisostomo.

**Formal analysis:** Michelle Ylade, Bipin Adhikari.

**Funding acquisition:** Jacqueline Deen.

**Investigation:** Michelle Ylade, Maria Vinna Crisostomo, Jedas Veronical Daag, Anna Maureen Cuachin, March Helena Lopez, Angela Macasero, Kristal An Agrupis, Jacqueline Deen.

**Methodology:** Michelle Ylade, Bipin Adhikari, Maria Vinna Crisostomo, Jedas Veronical Daag, Anna Maureen Cuachin, March Helena Lopez, Angela Macasero, Kristal An Agrupis, Jacqueline Deen.

**Project administration:** Michelle Ylade, Maria Vinna Crisostomo, Jedas Veronical Daag, Anna Maureen Cuachin, March Helena Lopez, Angela Macasero, Kristal An Agrupis.

**Resources:** Michelle Ylade.

**Supervision:** Jacqueline Deen.

**Validation:** Michelle Ylade, Jacqueline Deen.

**Visualization:** Michelle Ylade, Jacqueline Deen.

**Writing – original draft:** Bipin Adhikari.

**Writing – review & editing:** Michelle Ylade, Bipin Adhikari, Maria Vinna Crisostomo, Jedas Veronical Daag, Anna Maureen Cuachin, March Helena Lopez, Angela Macasero, Kristal An Agrupis, Jacqueline Deen.

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
