## [Decision Letter · Decision Letter 0]

22 Jul 2024

PGPH-D-24-01211

Impact of the COVID-19 pandemic and a supertyphoon on participants of a longitudinal cohort study: A quantitative study in Cebu, Philippines

Dear Dr. Adhikari,

Thank you for submitting your manuscript to PLOS Global Public Health. After careful consideration, we feel that it has merit but does not fully meet PLOS Global Public Health’s publication criteria as it currently stands. Therefore, we invite you to submit a revised version of the manuscript that addresses the points raised during the review process.

See below the the reviewer comments, and additional editorial comments and suggestions for addressing the key issues on the current form of the manuscript. Realising the sufficient data availability, the study can be improved scientifically in the line of method, analyses and inferential interpretation. Refer to my suggestions in the additional editorial comments below. Also thoroughly revise the manuscript for the language and presentation with point-by-point responses to the comments from reviewers and editor for further consideration. 

We look forward to receiving your revised manuscript.

Kind regards,

Sheikh Taslim Ali, M.Sc., Ph.D.

Academic Editor

Journal Requirements:

2. Please provide your detailed Financial Disclosure statement. This is published with the article. It must therefore be completed in full sentences and contain the exact wording you wish to be published.

**Please only choose the relevant sentences from below**

a. Please clarify all sources of funding (financial or material support) for your study. List the grants (with grant number) or organizations (with url) that supported your study, including funding received from your institution. 

b. State the initials, alongside each funding source, of each author to receive each grant.

c. State what role the funders took in the study. If the funders had no role in your study, please state: “The funders had no role in study design, data collection and analysis, decision to publish, or preparation of the manuscript.”

d. If any authors received a salary from any of your funders, please state which authors and which funders.

If you did not receive any funding for this study, please simply state: “The authors received no specific funding for this work.

3. Figure 1: please (a) provide a direct link to the base layer of the map (i.e., the country or region border shape) and ensure this is also included in the figure legend; and (b) provide a link to the terms of use / license information for the base layer image or shapefile. We cannot publish proprietary or copyrighted maps (e.g. Google Maps, Mapquest) and the terms of use for your map base layer must be compatible with our CC-BY 4.0 license. 

Additional Editor Comments (if provided):

Refer to the comments from the reviewers, they differ on the current form of your manuscript. Along with these, I like you to consider following key issues of the manuscript for major revision.

- The research questions are not clearly linked with the outcome variables of their interest. Define the outcome variables and link them with the research questions in their rationale, which not clear in the text.

- The quantitative analyses are only descriptive in nature merely just presented the data they have collected. No inferential quantification has been made on the impact assessment to make the result more scientific. Add such inferential analyses with uncertainty in the estimates.

- As author claimed the study design is a longitudinal cohort (until Oct 2022), but no temporal assessment has been done. Authors can consider the variation in outcome variables over time across the follow up and compare against baseline scenario in 2017.

- Not statistical analyses has been made to illustrate the individual and combined impact of two distinct disasters COVID-19 and super-typhoon specifically in the island country. Authors also could perform the risk analyses on these outcomes for these disasters and compare.

- Finally the language and presentation of the results have significant scope of improvement.

Along with addressing the reviewer comments, the above listed key issues should be incorporated in the revised manuscript for further consideration. I understand it will take a bit of time, lets us know if you need more time.

Reviewers' comments:

Reviewer's Responses to Questions

**Comments to the Author**

1. Does this manuscript meet PLOS Global Public Health’s publication criteria? Is the manuscript technically sound, and do the data support the conclusions? The manuscript must describe methodologically and ethically rigorous research with conclusions that are appropriately drawn based on the data presented.

Reviewer #1: Yes

Reviewer #2: Partly

Reviewer #3: Yes

2. Has the statistical analysis been performed appropriately and rigorously?

Reviewer #1: Yes

Reviewer #2: N/A

Reviewer #3: Yes

3. Have the authors made all data underlying the findings in their manuscript fully available (please refer to the Data Availability Statement at the start of the manuscript PDF file)?

Reviewer #1: Yes

Reviewer #2: Yes

Reviewer #3: Yes

4. Is the manuscript presented in an intelligible fashion and written in standard English?

Reviewer #1: Yes

Reviewer #2: Yes

Reviewer #3: Yes

5. Review Comments to the Author

Reviewer #1: The authors assessed the impact of the COVID-19 pandemic and a supertyphoon in Cebu, Philippines. I have some comments.

1. Title: "cohort study" should NOT be emphased, since this is a cross-sectional study.

2. The results are descriptive. The authors can do some statistical inference, e.g. 1. comparing the impact between COVID-19 pandemic and the supertyphoon; 2. assessing the influential factors of COVID-19 pandemic and the supertyphoon.

3. Page 3: pls replace "Covid-19" with "COVID-19", deleting the "USD" before "137.7 billion".

4. Section Participant: there is no need to describe vaccination.

5. Figure 3 and Figure 4: It is better to present the percentages additionally. And what are the differences between Figure 2 and Figure 3?

6. Table 1: pls delete "n(%)" and you can explain the meanings of figures in the footnote of the table. In addition, pls delete the colons in the first column.

7. Tables 2-4 are not professional.

8. Figure legends: pls replace "COVID" with "COVID-19".

Reviewer #2: I'm sorry to say that I don't think this manuscript has enough interesting findings or has advanced methodology innovation to be published.

Start from the title, it is very unclear what exactly the impact was of the author's primary research of interest. Reading through the full text dose not help to answer this question.

Secondly, why did the authors mention COVID-19 and typhoon together in the title? I was expecting to see if the authors used some proper statistical methods to find the joint impact of COVID-19 and typhoon on something, say participant's mental health status for example. But it seems the authors just mentioned these two separately. I couldn't find anything interesting enough.

Thirdly, I don't think this manuscript presented clear causal relationship investigation on either COVID-19 or typhoon's effect. More sophisticated quantitative methods are required.

Reviewer #3: The study aimed to investigate the effects of COVID-19 and Supertyphoon Rai on adolescent participants and their corresponding household head or parent/guardian. The research focused on the participants' financial situation, mental health, physical well-being, and social impact. The study is well-constructed, and here are some of my comments that I hope will help the authors enhance their manuscript.

1. The topic is about the impact of COVID-19 and Supertyphoon Rai. But in the “data collection section”, it says “supertyphoon Odette” in multiple places. Please clarify or correct it.

2. Some language needs to be a little more accurate.

1) “more than two-third” means more than 66.7%. But in the below two sentences, the proportions are 63.6% and 63.9%, respectively.

More than two-third of the respondents (1673/2360; 63.6%)

More than two-thirds of the respondents (1680/2360; 63.9%) reported

2) “A major proportion of the respondents (1101/2630; 41.9%) had moderate to severe impact on relationship with family members, relatives, friends and neighbors. ”

“Major proportion” normally means more than 50%. When the proportion is 41.9%, suggest using “A significant/large/considerable proportion of the respondents (41.9%)...”

3. Figures 2, 3, and 4 are very difficult to read and interpret. Suggest using bar chart.

6. PLOS authors have the option to publish the peer review history of their article (what does this mean?). If published, this will include your full peer review and any attached files.

**Do you want your identity to be public for this peer review?** For information about this choice, including consent withdrawal, please see our Privacy Policy.

Reviewer #1: No

Reviewer #2: No

Reviewer #3: No

---

## [Decision Letter · Decision Letter 1]

25 Sep 2024

PGPH-D-24-01211R1

Impact of the COVID-19 pandemic and a supertyphoon: A quantitative study in Cebu, Philippines

Dear Dr. Adhikari,

Thank you for submitting your manuscript to PLOS Global Public Health. After careful consideration, we feel that it has merit but does not fully meet PLOS Global Public Health’s publication criteria as it currently stands. Therefore, we invite you to submit a revised version of the manuscript that addresses the points raised during the review process.

Please refer to the **Additional Editor Comments **below carefully.

We look forward to receiving your revised manuscript.

Kind regards,

Sheikh Taslim Ali, M.Sc., Ph.D.

Academic Editor

Journal Requirements:

Additional Editor Comments (if provided):

I was expecting authors should take the comments seriously while revising their manuscripts. I found authors didn't address several key issues raised. I am still optimistic that authors could address these comments by incorporating additional simple analyses quickly before am able to have a decision on it. Along with reviewers comments, I have following comments to address.

1. Initial Comment: - The research questions are not clearly linked with the outcome variables of their interest. Define the outcome variables and link them with the research questions in their rationale, which not clear in the text.

AUTHORS' Respond: Thank you for the suggestion. We have added research question under the section ‘Data collection’ to clarify how the research question is related to the outcome variable.

Comment on Revised version: I suggest you can briefly utter those outcome variable with their rationale in the last paragraph of introduction for general reader first. Then for each outcome, you can link in the data collection section as you have presented now.

2. Initial Comment:- The quantitative analyses are only descriptive in nature merely just presented the data they have collected. No inferential quantification has been made on the impact assessment to make the result more scientific. Add such inferential analyses with uncertainty in the estimates.

AUTHORS' Respond: Thank you for the suggestion. Our original statistical analysis plan only intended to present the descriptive nature of the impacts by COVID-19 and typhoon Odette. Based on your suggestion, we have added the inferential analysis, in new tables (1-10). The descriptive (original) tables are now added as supplementary files.

Comment on Revised version:

- First, do you think all 10 tables (table 1 to 10) should be the main text figure? I suggest select maximum 2-4 key table for main text and rest should be in supplementary or try to combine these tables in case. I am confused with the table numbering only two tables (table S1 and S2) are cited in the main text but they listed 4 tables as supporting information (page 15) with typo in their labelling. Make sure all tables and figures should be discusses and cited in the main text.

- Secondly, including p-value in the table looks better. I understand authors performed several tests including Chi-squared tests, Fisher’s exact and the Mann−Whitney U test based on the data. I suggest to provide the details of the test used with rationale in foot notes of each table.

3. Initial Comment:- As author claimed the study design is a longitudinal cohort (until Oct 2022), but no temporal assessment has been done. Authors can consider the variation in outcome variables over time across the follow up and compare against baseline scenario in 2017.

AUTHORS' Respond: Thank you for the suggestion. Since this study surveyed the participants of a longitudinal cohort study, we thought to keep this information in the title and at various section, where feasible. But we agree with you, it may have added unnecessary confusion, so we have deleted the terms and have shortened it where feasible. For instance, Revised title: Impact of the COVID-19 pandemic and a supertyphoon: A quantitative study in Cebu, Philippines.

Comment on Revised version: It is not only the question of title, my suggestion to see the outcome status temporarily (under each follow ups) where the baseline scenario can be of 2017. As the study was conducted longitudinally, why not considering the temporal analyses on the outcomes using simply the consecutive followup data. I am not convinced with the author response and it is not cross-sectional study.

4. Initial Comment:- No statistical analyses has been made to illustrate the individual and combined impact of two distinct disasters COVID-19 and super-typhoon specifically in the island country. Authors also could perform the risk analyses on these outcomes for these disasters and compare.

AUTHORS' Respond: Thank you for the suggestion. We have added the inferential statistical analyses in the revised manuscript (tables 1-10)

Comment on Revised version: Here I meant to quantify the risk. Tables 1-10 could help to identify the risk factors, but any way to quantify them, which could be helpful in the line of interventions.

5. Fig 1 is missing in revised manuscript, is it by mistake?

6. The Fig 2-4, revise the x-lable as "Number of participants (%)" and can be combined in one figure for better integrated comprehension.

7. Finally, the text should be revised for English and presentation (for general reader).

Reviewers' comments:

Reviewer's Responses to Questions

**Comments to the Author**

1. If the authors have adequately addressed your comments raised in a previous round of review and you feel that this manuscript is now acceptable for publication, you may indicate that here to bypass the “Comments to the Author” section, enter your conflict of interest statement in the “Confidential to Editor” section, and submit your "Accept" recommendation.

Reviewer #1: All comments have been addressed

Reviewer #3: All comments have been addressed

2. Does this manuscript meet PLOS Global Public Health’s publication criteria? Is the manuscript technically sound, and do the data support the conclusions? The manuscript must describe methodologically and ethically rigorous research with conclusions that are appropriately drawn based on the data presented.

Reviewer #1: Yes

Reviewer #3: Yes

3. Has the statistical analysis been performed appropriately and rigorously?

Reviewer #1: Yes

Reviewer #3: Yes

4. Have the authors made all data underlying the findings in their manuscript fully available (please refer to the Data Availability Statement at the start of the manuscript PDF file)?

Reviewer #1: Yes

Reviewer #3: Yes

5. Is the manuscript presented in an intelligible fashion and written in standard English?

Reviewer #1: Yes

Reviewer #3: Yes

6. Review Comments to the Author

Reviewer #1: The authors have addressed most of my comments. I have a suggestion on how to show the table. For example, in Table 1, the number of participants should NOT be placed in a separate cell, instead, you can write "Financially impacted (n=2,486)" in a single cell. In addition, you can explain what do the number outside and in brackets mean in the footnote of the table.

Reviewer #3: (No Response)

7. PLOS authors have the option to publish the peer review history of their article (what does this mean?). If published, this will include your full peer review and any attached files.

**Do you want your identity to be public for this peer review?** For information about this choice, including consent withdrawal, please see our Privacy Policy.

Reviewer #1: No

Reviewer #3: No

---

## [Editor Report · Decision Letter 2]

13 Nov 2024

Impact of the COVID-19 pandemic and a supertyphoon: A quantitative study in Cebu, Philippines

PGPH-D-24-01211R2

Dear Dr. Adhikari,

We are pleased to inform you that your manuscript 'Impact of the COVID-19 pandemic and a supertyphoon: A quantitative study in Cebu, Philippines' has been provisionally accepted for publication in PLOS Global Public Health.

Best regards,

Sheikh Taslim Ali, M.Sc., Ph.D.

Academic Editor

Thank you for the the revision.

I can see some of the minor issues are still need to be addressed, which should be done during the proof editing.

1. Authors are now acknowledging that the study is not a longitudinal cohort as a whole, rather a cross-sectional. But in the text (Abstract and Study Context sections), it is till emphasised as if the study is a longitudinal cohort. Revise the text accordingly.

2. Revise labels of Fig 2 as 'Number of participants (%)' instead of 'No of participants (%)'.